# Overnutrition during Pregnancy and Lactation Induces Gender-Dependent Dysmetabolism in the Offspring Accompanied by Heightened Stress and Anxiety

**DOI:** 10.3390/nu16010067

**Published:** 2023-12-25

**Authors:** Gonçalo M. Melo, Adriana M. Capucho, Joana F. Sacramento, José Ponce-de-Leão, Marcos V. Fernandes, Inês F. Almeida, Fátima O. Martins, Silvia V. Conde

**Affiliations:** iNOVA4Health, NOVA Medical School, Faculdade de Ciências Médicas, Universidade NOVA de Lisboa, Rua Camara Pestana, 6, Edificio 2, 1150-082 Lisboa, Portugal; goncalo.melo@nms.unl.pt (G.M.M.); adriana.capucho@nms.unl.pt (A.M.C.); joana.sacramento@nms.unl.pt (J.F.S.); jose.poncedeleao@nms.unl.pt (J.P.-d.-L.); viniciusmar3@gmail.com (M.V.F.); ines.almeida@nms.unl.pt (I.F.A.); fatima.martins@nms.unl.pt (F.O.M.)

**Keywords:** overnutrition, offspring, dysmetabolism, metabolic diseases, stress, anxiety

## Abstract

Maternal obesity and gestational diabetes predispose the next generation to metabolic disturbances. Moreover, the lactation phase also stands as a critical phase for metabolic programming. Nevertheless, the precise mechanisms originating these changes remain unclear. Here, we investigate the consequences of a maternal lipid-rich diet during gestation and lactation and its impact on metabolism and behavior in the offspring. Two experimental groups of Wistar female rats were used: a control group (NC) that was fed a standard diet during the gestation and lactation periods and an overnutrition group that was fed a high-fat diet (HF, 60% lipid-rich) during the same phases. The offspring were analyzed at postnatal days 21 and 28 and at 2 months old (PD21, PD28, and PD60) for their metabolic profiles (weight, fasting glycemia insulin sensitivity, and glucose tolerance) and euthanized for brain collection to evaluate metabolism and inflammation in the hypothalamus, hippocampus, and prefrontal cortex using Western blot markers of synaptic dynamics. At 2 months old, behavioral tests for anxiety, stress, cognition, and food habits were conducted. We observed that the female offspring born from HF mothers exhibited increased weight gain and decreased glucose tolerance that attenuated with age. In the offspring males, weight gain increased at P21 and worsened with age, while glucose tolerance remained unchanged. The offspring of the HF mothers exhibited elevated levels of anxiety and stress during behavioral tests, displaying decreased predisposition for curiosity compared to the NC group. In addition, the offspring from mothers with HF showed increased food consumption and a lower tendency towards food-related aggression. We conclude that exposure to an HF diet during pregnancy and lactation induces dysmetabolism in the offspring and is accompanied by heightened stress and anxiety. There was sexual dimorphism in the metabolic traits but not behavioral phenotypes.

## 1. Introduction

Metabolic syndrome (MS) is a serious public health problem affecting about 25% of the general population worldwide [1]. The definition of this syndrome has recently evolved to include a group of at least three of the five cardio-metabolic abnormalities, including high blood pressure, central obesity, insulin resistance (IR), elevated blood triglycerides, and atherogenic dyslipidemia, which, together, lead to an increased risk of cardio-metabolic pathologies [1]. The presence of MS-related disorders also has an impact on the central nervous system, causing neurological and neurodegenerative diseases [2]. The intricate relationship between dysmetabolism and neurodegeneration has been described to be relevant not only to aging-associated comorbidities but also in mild cognitive impairment in younger ages and during the initial steps of an organism’s development. In fact, it is known that the nutritional status of the mother can prematurely influence the onset of diseases in the offspring, and there is significant research focused on understanding its impact on the metabolic status and nervous system of the offspring [3]. Studies in animals have shown that maternal obesity can increase the likelihood of metabolic and neurodevelopmental diseases in the offspring, such as cognitive impairment, autism spectrum disorders, and attention deficit hyperactivity disorder [2,3,4]. Moreover, maternal obesity and dysmetabolism are linked to a heightened risk of psychiatric disorders in the offspring, encompassing conditions such as anxiety, depression, schizophrenia, psychosis, eating disorders, and food addiction [5]. Several mechanisms have been postulated to underlie these neurodevelopment and psychiatric conditions, including oxidative stress and inflammation-induced malprogramming; the deregulation of insulin, glucose, and leptin signaling; the dysregulation of synaptic transmission; or even the deregulation of the gut–brain axis [6,7]. Nevertheless, the precise ones are not completely understood.

Recent evidence indicates that many components of a mother’s diet are pivotal in shaping aspects of the offspring’s health, including the microbiome and the neonatal immune system [8]. Research suggests that the maternal diet can influence the development of the offspring’s brain, endocrine system, and long-term behavior [9]. Therefore, maintaining a balanced and nutritious diet during pregnancy is crucial to meet the increased demands on the mother’s body, as adequate nutrient intake is essential to support the offspring’s growth and ensure a healthy birth weight.

During the perinatal period, maternal obesity can elevate the risk of gestational diabetes and hypertension, potentially affecting placental function and fetal energy metabolism [10]. Obesity during pregnancy is also associated with complex alterations in the neuroendocrine, metabolic, and inflammatory processes, which can potentially influence fetal hormonal exposure and nutrient supply [11,12,13].

Throughout pregnancy, the mother’s body undergoes significant metabolic changes, which, if left untreated, can contribute to certain health disparities. In early pregnancy, there is an increase in maternal insulin secretion to facilitate the transportation of glucose and amino acids to the developing fetus [12,13]. However, this increased insulin secretion, combined with high levels of circulating lipids and the consequent increase in visceral adiposity, often results in the development of insulin resistance in maternal cells [10,11,12,13]. When lactation begins, a metabolic shift occurs, redirecting resources from storage milk synthesis, with this process being intricately regulated by lactogenic hormones such as insulin, prolactin, and glucocorticoids, as well as by cytokines, growth factors, and substrate availability [14]. The alterations promoted by lactation causes several positive changes in the metabolic processes. This leads to better glucose regulation, reducing insulin production, improving insulin sensitivity, and decreasing β-cell proliferation. In addition, lipid metabolism becomes less active in certain tissues, and stored lipids are mobilized to facilitate the transport of lipids to the mammary gland for milk production [11,13,14]. As a result of these metabolic changes during lactation, there is a reduction in postpartum adiposity, potentially decreasing the risk of obesity. Based on this evidence, Stuebe and Rich-Edwards proposed the “reset hypothesis”, which suggests that lactation plays a crucial role in resetting the metabolic processes that take place during labor [15].

However, when maternal overnutrition continues, the offspring’s metabolism as well as neurological development may be impacted. Determining the mechanisms behind the expected impact of hypercaloric diet-fed mothers on the offspring’s metabolism and neurological development is the main goal of this study. Additionally, since it is well known that sex influences both metabolic [16] and neurodegenerative disorders [17], development, and progression, we will also focus our attention on the differences of the impact of maternal overnutrition on male vs. female offspring metabolism and stress and anxiety behaviors.

## 2. Materials and Methods

### 2.1. Animals

Experiments were performed on 12-week-old female and 24-week-old male Wistar rats (200–400 g), obtained from Charles River Laboratories (Barcelona, Spain) and maintained at the NOVA Medical School animal facility. Animals were kept under temperature and humidity control (21 ± 1 °C; 55 ± 10% humidity) and a regular light (08.00–20.00 h) and dark (20.00–08.00 h) cycle, with food and water provided ad libitum. After randomization, animals were divided into a normal chow diet group (NC) or a high-fat (HF) diet group. The NC group was fed a standard diet (7.4% fat + 75% carbohydrates (4% sugar) + 17% protein, SDS diets RM1, Probiológica, (Lisboa, Portugal), and the HF diet group was fed a 60% lipid-rich diet (61.6% fat + 20.3% carbohydrate + 19.1% protein, TestDiet, St. Louis, MO, USA). After confirming that the females were pregnant, they stayed in cages, with 1 animal per cage. During the breastfeeding period, the pup’s body weight was weekly motorized. On the weaning day—21 days postnatal (PD21)—both male and female offspring were separated from the mothers and kept in cages, with 4 animals per cage. They were randomly divided, and part of them were sacrificed at PD21 and 28 days postnatal (PD28), and the other group was kept alive until 45 days postnatal, where metabolic and behavioral parameters were assessed. Metabolic evaluations were performed in the mothers by using the insulin tolerance test (ITT) and oral glucose tolerance test (OGTT) (Figure 1).

Pups were also tested for insulin sensitivity through the ITT and for glucose tolerance through the OGTT. Behavior was assessed by open field, Y maze, elevated plus maze (EPM), light dark box, sucrose competition, food competition, and novel object recognition (NOR) tests. At the time of sacrifice, PD21, PD28, and PD60, the animals were anesthetized and sacrificed with pentobarbital (60 mg/kg i.p.), and the brains were dissected and frozen at −80 °C for further protein analysis.

All experiments and animal care were performed in accordance with the European Union Directive for Protection of Vertebrates Used for Experimental and Other Scientific Ends (2010/63/EU) and with the ARRIVE guidelines. Experimental protocols were approved in 2021 by the Ethics Committee of NOVA Medical School/Faculdade de Ciências Médicas (no. 94/2021/CEFCM) and by the Portuguese Authority for Animal Health (DGAV, Ref no. 0421/000/000/2021).

### 2.2. Metabolic Evaluation

#### 2.2.1. Intravenous Insulin Tolerance Test (ITT)

Insulin sensitivity was assessed in conscious mothers using the intravenous ITT, which provides an estimate of overall insulin sensitivity [18]. In brief, following an overnight fast, an insulin bolus (0.1 U/Kg, Humulin Regular, Lilly, Alcobendas, Spain) was administered in the tail vein, and the subsequent 15 min decline in blood sugar levels was measured. Blood samples were collected by tail tipping, and glycemia was assessed using a glucometer (Precision Xtra Meter, Abbott Diabetes Care, Amadora, Portugal) and test strips (Abbott Diabetes Care, Amadora, Portugal). The constant rate of decline in plasma glucose (K_ITT_) was calculated as previously outlined [19].

#### 2.2.2. Intraperitoneal Insulin Tolerance Test (ipITT)

In the offspring, insulin sensitivity was assessed using the intraperitoneal ITT. Pups were fasted for approximately 6 h with free access to water. Basal blood glucose was measured, and 0.1 U/Kg of insulin (Humulin Regular, Lilly, Alcobendas, Spain) was administered intraperitoneally. Glucose was measured from blood collected from the tip of the tail vein at 5, 10, 15, 30, 45, 60, and 120 min after the injection with a glucometer (Precision Xtra Meter, Abbott Diabetes Care, Amadora, Portugal) and test strips (Abbott Diabetes Care, Amadora, Portugal). Glucose excursion curves from the ITT were used to calculate the area under the curve and therefore to evaluate insulin sensitivity.

#### 2.2.3. Oral Glucose Tolerance Test (OGTT)

Glucose tolerance was assessed through the OGTT. Both mothers and pups that were fasted overnight for approximately 12–15 h were administered with a glucose solution (2 g/kg in a 10 uL/g body weight volume) by gavage after the measurement of basal glycemia. Glucose levels were measured at 0 and 15, 30, 60, and 120 min after the oral gavage by tail tipping using a glucometer (Precision Xtra Meter, Abbott Diabetes Care, Amadora, Portugal) and test strips (Abbott Diabetes Care, Amadora, Portugal). The glucose excursion curves were used to calculate the area under the curve.

### 2.3. Behavioral Assessment

#### 2.3.1. Open Field (OF)

The OF test is used to assess gross motor activity, anxiety, and willingness to explore [20]. In this test, animals were placed for 5 min in a square arena (70 cm × 70 cm × 75 cm). Their behaviors were recorded at 30 frames per second and analyzed using Bonsai software (version 7.0). For that, an inner and central zone in the maze was defined (40 cm × 40 cm), and the following were measured: the total distance covered, distance covered in the inner zone, total immobility time, immobility time in the inner zone, average velocity, and average velocity in the inner zone. It was considered that the animal was in the inner zone when more than half of its body was in the 40 cm area. Immobility was considered when the animal was stationary. Data presented are results from a single trial for each rat.

#### 2.3.2. Y Maze

To assess short-term spatial memory, animals were challenged to a Y-shaped maze with 120° between each arm with a 10 cm width and 30 cm height. First, animals were submitted to a training session in which a rat was placed in the start point (A arm) of the Y maze with a closed arm and allowed to freely explore it for 5 min. The experimental session occurred 1 h after training where the “novel” arm was open. The animal freely explored the maze for 5 min. The exploratory capacity of the animal was evaluated as well as the time spent both in the novel arm and in the arm that was always open. The first arm choice when both arms were open was evaluated, as well as the number of triads (i.e., ABC, CAB, or BCA) and entries. The spontaneous alternative behavior score (%) for each rat was calculated as the ratio of the number of alternations to the possible number (total number of arm entries minus two) multiplied by 100. Videos were recorded as explained in Section 2.4.1.

#### 2.3.3. Elevated Plus Maze (EPM)

EPM test is used to characterize anxiety-related behavior. The apparatus is composed of a small central platform with four arms angled 90° from each other radiating outwards, and it is raised above the ground to a height of 112 cm; it is 50 cm in length and 12 cm in width. Alternating arms are enclosed by high opaque walls of 41.5 cm in height with open tops. Animals were placed in the center of the maze and allowed to freely explore for 5 min for a single trial and recorded using an overhead camera. The time spent on open arms and the number of entries were measured. Data presented are results from a single trial for each rat. Videos were recorded and analyzed as described in Section 2.3.1.

#### 2.3.4. Light Dark Box (LDB)

The LDB test is used to evaluate anxiety responses in rodents. The apparatus is composed of two compartments. The large light compartment (2/3 of the box) is brightly lit and open, while the small dark compartment (1/3 of the box) is covered and dark. These two compartments are connected by a door of 10 cm. Rats were placed in the corner of the brightly illuminated chamber and were allowed to freely explore the maze for 5 min. Parameters such as the time spent in the light/dark compartment and the number of transitions were evaluated. Data presented are results from a single trial for each rat. Videos were recorded and analyzed as described in Section 2.3.1.

#### 2.3.5. Novel Object Recognition (NOR)

NOR test is commonly used in rodents to evaluate cognition, particularly recognition memory. This test was performed in the same maze as the OF test, with two different objects. The objects were different in shape and appearance. This test is composed of a train where the animals are allowed to explore the maze with two identical objects placed at an equal distance. One hour later, rats are allowed to explore the arena in the presence of the familiar object and a novel object to test long-term recognition memory. The time spent exploring each object and the ratio between the novel/novel+ familiar (interaction time) were analyzed. Data presented are results from a single trial for each rat. Videos were recorded and analyzed as described in Section 2.3.1.

#### 2.3.6. Block Test

Block test is used to assess olfactory function. This test evaluates sensitivity to social smells, which is an ability in rats [21]. Housed animals were exposed to five wood blocks (7 cm × 2 cm × 2 cm) (Ultragene, Santa Comba Dão, Portugal) placed inside each cage for 1 week. Each rat underwent two training sessions. In the trial test, one block was replaced by a block that was originally in a cage with a different set of animals. Rat was videotaped for 1 min. Videos were recorded and analyzed as described in Section 2.3.1. Parameters such as the time that animals take to recognize the novel block and the time spent to sniff the novel block were evaluated. The animal was only considered to be sniffing the block when its nose was touching/very close to the block. Data presented are results from a single trial for each rat.

#### 2.3.7. Food Competition

Food competition is a test that not only measures the antagonistic and dominance behaviors of offspring but also assesses voracity and food preferences. The animals were fasted for 24 h. Following the fasting period, the animals were randomly paired with a same-sex partner and placed in the maze (44 cm wide, 49 cm long, and 20 cm high box). Inside this maze, on one of its sides, there was an integrated dispenser with a retractable door, where the food was stored. The pair was placed inside the maze, and for 2 min, the door was kept closed. After 2 min, the dispenser’s door was opened to allow the rats access to the food. During the following 8 min, observations were made regarding the amount of food consumed, the occurrence of antagonistic behaviors, their duration, as well as the frequency of grooming behaviors exhibited by the rats. Data presented are results from a single trial for each pair. 

#### 2.3.8. Water/Sucrose Competition

Similarly, to the food competition, the water/sucrose competition measures antagonistic and dominance behaviors in offspring, as well as their liquid consumption preferences. 

The animals were subjected to a 24 h liquid deprivation. During this period, only food was available ad libitum. After the fasting period, the animals were randomly paired as described in Section 2.3.7. Inside this maze, on one of its faces, there was a dispenser set at a 45° angle capable of holding a water bottle. This dispenser featured a removable door, controlling the rats’ access to the liquid. The dispenser closed for 2 min. After that, for 8 min, the bottles became available, and the rats had access to it. Observations were made regarding the liquid intake, the occurrence of antagonistic behaviors, their duration, as well as the frequency of grooming behaviors exhibited by the rats. 

The antagonistic behaviors included stealing pellets from another individual’s mouth or paws (food competition), pushing the other’s head while drinking from the water bottle (water/sucrose competition), and aggression, such as biting, moving the other individual away, or just fighting.

The data presented are results from a single trial for each pair. After the water trial, the same pair was exposed to the sucrose trial with the same characteristics, but the sucrose level was 1 g/L. 

### 2.4. Ex Vivo Analysis 

After 21, 28, and 60 days, animals were sacrificed, and certain brain areas—frontal cortex, hypothalamus, and hippocampus—were dissected and harvested for protein analysis.

#### 2.4.1. Tissue Lysate Preparation and Western Blot Analysis

For Western blot analysis, brain samples (20 mg) were homogenized in RIPA buffer (50 mM Tris-HCl pH 7.4, 150 mM NaCl, 2 mM EDTA, 0.1% SDS, 0.25% sodium deoxycholate) and sonicated 3 times in cycle of 30 s. Protein extracts were centrifuged for 10 min at 13,000 rpm at 4 °C, and the supernatant was collected. 

Proteins (8 μg) were separated by electrophoresis in 10% sodium dodecylsulfate-polyacrylamide gel electrophoresis (SDS-PAGE), followed by transference to a nitrocellulose membrane (BioRad, Dusseldorf, Germany). After transference, the membrane was incubated with blocking solution composed of milk or bovine serum albumin (BSA) at room temperature for 1 h. Primary antibody incubations were carried out overnight at 4 °C using the following concentrations: phosphorylated insulin receptor (anti-IR-p, rabbit 1:500, abcam, Cambridge, UK); total insulin receptor (anti-IR T, mouse 1:1000, Santa Cruz Biotechnology, Heidelberg, Germany); phosphorylated mitogen-activated protein kinase (AMPK) (anti-pAMPK 1:500 rabbit, Cell Signaling, Beverly, MA, USA) and total AMPK (anti-AMPK rabbit 1:1000, Cell Signaling, Beverly, MA, USA); synaptosome-associated protein (SNAP25) (anti-SNAP25 1:500, Santa Cruz Biotechnology, Heidelberg, Germany); postsynaptic density protein (PSD95) (anti-PSD95 1:1000, Invitrogen, Waltham, MA, USA); vesicular glutamate transporter 1 (vGLuT) (anti-vGLuT 1:1000, MilliporeSigma, Burlington, MA, USA); glial fibrillary acidic protein (GFAP) (anti-GFAP 1:2000, Palex, Oeiras, Portugal); and tumor necrosis factor alpha (TNF-α) (anti-TNF-α 1:200, Sicgen, Cantanhede, Portugal). Secondary antibody was incubated at room temperature for 1.5 h. Detection procedures were carried out according to ECL system (GE Healthcare Life Sciences; Little Chalfont, UK), and the signal was detected using a ChemiDocTM Imaging System (Bio-Rad, Hercules, CA, USA). Membranes were re-probed for the β-actin (anti-β-actin 1:2000, Sicgen, Cantanhede, Portugal).

### 2.5. Data Analysis

Data were analyzed using GraphPad Prism Software, version 8.0.2 (GraphPad Software Inc., San Diego, CA, USA), and presented as mean values with the standard error of the mean (SEM). The significance of the differences between the groups was calculated using Student’s *t* test and one- and two-way ANOVA with Dunnett’s and Bonferroni and Sidak’s multiple comparison tests. Differences were considered significant at *p* ≤ 0.05. Experimental groups constituted 8–10 animals.

## 3. Results

### 3.1. Effect of Overnutrition during Pregnancy and Lactation on Metabolic Parameters in the Mothers

Mothers that were overnourished for 6 weeks under an HF diet were evaluated for body weight and glucose metabolism, assessed using basal glycemia and OGTT; insulin sensitivity, evaluated using the ipITT; and body composition, assessed using the liver and AT depots’ weight measurements.

As expected, feeding an HF diet to the mothers for 6 weeks promoted an increase in basal glycemia (basal glycemia: NC mothers = 69.14 ± 1.792; HF mothers = 78.88 ± 3.573, mg/dL, *p* < 0.05) and a drastic decrease in insulin sensitivity (k_ITT_: NC mothers = 4.864 ± 0.3957; HF mothers = 1.849 ± 0.3188, % glucose/min, *p* < 0.0001) without significantly affecting glucose tolerance (Table 1). Moreover, the HF diet did not alter the body weights or liver and AT depots’ weights in the mothers, which can be attributed to the pregnancy and offspring delivery during that period (Table 1).

### 3.2. Effect of Overnutrition during Pregnancy and Lactation on Metabolic Function in the Offspring

#### 3.2.1. Insulin Action and Glucose Homeostasis

Female and male offspring born from overnourished mothers during pregnancy and lactation periods were evaluated for body weight, basal glycemia, insulin sensitivity, and glucose tolerance (Figure 2).

HF feeding to mothers promoted a significant increase in the body weights of both the female and male descendants during all the postnatal periods analyzed (Figure 2A). The body weights of the female offspring from the HF mothers increased by 65.1%, 33.3%, and 9.6% at PD 21, 28, and 60, respectively (Figure 2A, left panel). The male offspring born from the overnourished mothers showed significant body weight increases of 63.9%, 45.6%, and 20.5% at PD 21, 28, and 60, respectively (Figure 2A, right panel).

Maternal overnutrition during pregnancy and lactation significantly increased the basal glycemia of the female offspring at PD21 and 28 in comparison to the female offspring from the NC mothers (basal glycemia: NC PD21 = 74.5 ± 6.5, HF PD21 = 95.6 ± 7.8, *p* < 0.01; NC PD28 = 61.0 ± 2.4, HF PD28 = 78.8 ± 2.6, *p* < 0.05), and the basal glycemia did not change at PD60 (Figure 2B, left panel). Interestingly, we found that during litter development, the NC female offspring showed an increase in basal glycemia that was indeed significant at PD60 (basal glycemia: NC PD21 = 81.3 ± 2.6, *p* < 0.01) when compared to the PD28 values, and in contrast, the HF female offspring had decreased basal glycemia from PD21 to PD28 (*p* < 0.05), and it was maintained at PD60 (basal glycemia: HF PD60 = 85.6 ± 2.0, *p* < 0.01). Basal glycemia was not altered by HF maternal feeding in male offspring in the period of development that was analyzed (Figure 2B, right panel).

Figure 2C,D represent the curves of glycemia during the ipITT performed on the offspring (Figure 2C) and the correspondent AUC of those curves (Figure 2D). From the curves, it can be seen that at PD21, the male offspring from the HF mothers presented a significant increase in glycemia levels throughout all of the ipITTs (Figure 2C, left panel), demonstrating a decrease in insulin sensitivity. At PD28, neither HF maternal feeding nor sex impacted the glycemia profile during the ipITT (Figure 2C, central panel). At PD60, the glycemia profiles during the ipITT showed that the male offspring had a significant increase in glycemia throughout the test, independently to the mothers’ diets during pregnancy and lactation (Figure 2C, right panel). Through the analysis of the glycemia AUC from the ipITT during offspring development in the initial 60 days, insulin sensitivity significantly increased in the female descendants of the NC mothers (AUC glycemia during ipITT: NC PD21 = 900.4 ± 36.2; NC PD28 = 677.2 ± 41.6, *p* < 0.01; NC PD60 = 724.8 ± 22.5, *p* < 0.05) (Figure 2D, left panel). In the male offspring, the glycemia AUC during the ipITT decreased from PD21 to PD28 and increased from PD28 to PD60 in both the NC and HF descendants (AUC glycemia during ipITT: NC PD21 = 807.8 ± 66.4; NC PD28 = 677.6 ± 23.9, *p* < 0.01; NC PD60 = 862.6 ± 19.5, *p* < 0.05; HF PD21 = 1022.2 ± 81.6; HF PD28 = 663.9 ± 41.2, *p* < 0.01; HF PD60 = 878.6 ± 34.9, *p* < 0.05) (Figure 2D, right panel), showing a significant increase in insulin sensitivity from PD21 to PD28 and a significant decrease in insulin sensitivity from PD28 to PD60 independent of maternal diet.

Finally, the impacts of maternal diet and offspring sex on the descendants’ glucose tolerance was evaluated using the OGTT (Figure 2E,F). The glycemia profiles showed that neither the mothers’ diets nor offspring sex impacted glucose tolerance at each PD that was evaluated (Figure 2E). However, the female offspring from the HF mothers showed a significant decrease in glucose tolerance in comparison to the female offspring of the NC mothers at PD21 (AUC glycemia during OGTT: NC PD21 = 19,955.2 ± 1033.4; HF PD21 = 23,999.4 ± 857.5; *p* < 0.05), an effect that is attenuated during offspring development (AUC glycemia during OGTT: HF PD21 = 23,999.4 ± 857.5; HF PD28 = 20,866.1 ± 345.6; HF PD60 = 20,076.8 ± 598.9; *p* < 0.05) (Figure 2F, left panel). Male offspring glucose tolerance was not altered by the diet fed to the mothers during development (Figure 2F, right panel). 

#### 3.2.2. Liver and Adipose Tissue Depots’ Weights

In Table 2 are the liver and adipose tissue depots’ weights of descendants of mothers submitted to a NC or HF diet during the gestation and lactation periods. Maternal overnutrition during pregnancy and lactation impacted both the male and female offspring AT depots’ weights mainly at PD60.

In the female offspring, overnutrition in mothers significantly increased visceral and perinephric AT at PD60 (visceral AT weight: NC females PD60 = 0.406 ± 0.038; HF females PD60 = 0.527 ± 0.055, *p* < 0.001; perinephric AT weight: NC females PD60 = 0.196 ± 0.017; HF females PD60 = 0.374 ± 0.068, *p* < 0.01, mg).

Regarding the male offspring, maternal overnutrition promoted an increase in liver weight at PD21 (NC males PD21 = 2.286 ± 0.086; HF males PD21 = 2.717 ± 0.126, *p* < 0.01) and at PD60 (NC males PD60 = 6.377 ± 0.326; HF males PD60 = 7.218 ± 0.465, *p* < 0.05). Also, an HF diet in mothers during pregnancy and lactation promoted an increase in the weight of visceral AT (NC males PD60 = 1.216 ± 0.182; HF males PD60 = 1.864 ± 0.152, *p* < 0.01) and genital AT (NC males PD60 = 2.324 ± 0.375; HF males PD60 = 3.642 ± 0.503, *p* < 0.01) in male descendants at PD60.

The brown AT weight was not altered significantly in neither the female nor male descendants of the HF mothers, at least until PD60. 

### 3.3. Effect of Overnutrition during Pregnancy and Lactation on Behavior Phenotype in the Offspring

In order to assess the impact of overnutrition during pregnancy and lactation on the offspring, a series of behavioral tests were conducted on the offspring at PD60. These tests were categorized into three main groups, presented in the following sections: Section 3.3.1. Anxiety and Stress; Section 3.3.2. Memory and Learning; and Section 3.3.3. Food/Drink Behavior.

#### 3.3.1. Anxiety and Stress

In this battery of tests (Figure 3), we incorporated the OF, EPM, and the LDB tests, which are all designed to measure anxiety and stress levels in the offspring [22].

Regarding the OF results, significant differences (*p* < 0.05) were evident among the female offspring groups concerning the distance covered within the inner zone. This suggests that the female offspring from mothers on an HF diet spent notably less time in the inner zone compared to those from the control mothers. However, no significant differences were observed among the other groups for the remaining three parameters: total distance, total immobility/total test time, and immobility within the inner zone/total test time (Figure 3A).

Concerning the results of the elevated plus maze (EPM) test, there were no significant differences detected between the groups (NC vs. HF) for the five parameters assessed: ratio of open arm/open + closed arm, ratio of closed arm/open + closed arm, number of poops, number of pees, and the frequency of alternations between arms (Figure 3B).

Finally, when it comes to the outcomes of the light–dark box test, there were no significant differences observed between the groups (NC vs. HF) in the three parameters analyzed: the proportion of time spent in the light zone relative to the total test time, the proportion of time spent in the dark zone relative to the total test time, and the number of alternations (Figure 3C).

#### 3.3.2. Memory and Learning

For this set of assessments (Figure 4), we used the Y maze, the novel object recognition (NOR), and the block test, designed to assess memory and learning [22].

The Y maze test was used to evaluate spatial learning and memory capacity in the offspring of both the NC and HF animals. There were no significant differences observed in the four parameters analyzed: the time the animals spent in the new arm, the ratio of time spent in the novel arm to the sum of time spent in both the novel and home arms, the number of entries into the novel arm, and alternative behaviors. The last parameter calculates the ratio of the number of alternations to the possible number (total number of arm entries minus two) (Figure 4A). 

Regarding the NOR results, there were no significant differences observed between the NC and HF groups in the three parameters measured: the time required to identify the new object, the ratio of time spent interacting with the novel object over the total time spent interacting with both the novel and familiar objects, and the ratio of time spent interacting with the familiar object over the total time spent interacting with both the novel and familiar objects (Figure 4B).

Finally, the animals’ olfactory working memories were assessed using the block test that assesses the ability of the animals to discriminate between familiar and novel, innocuous scents [21]. The results are presented in Figure 3C. No significant differences were observed between the experimental groups for the three parameters analyzed: the ratio of time spent sniffing the novel block over the total time spent sniffing both the novel and familiar blocks, the ratio of time spent sniffing the familiar block over the total time spent sniffing both the novel and familiar blocks, and the time required to identify the new block. However, it is worth noting that there is a noticeable trend in the last parameter, where the offspring of the HF animals tend to take more time to complete this task.

#### 3.3.3. Food/Drink Behavior

To assess the competition for food and drink, the respective two tests were used: food competition and water/sucrose competition (Figure 5).

The food competition test was used to assess both the competition for food, the voracity of feeding, as well as the dominance in the offspring in both the NC and HF animals. No significant differences were observed in any of the five analyzed parameters: time spent eating, food intake, antagonistic behavior, time spent displaying antagonistic behavior, and grooming. However, a trend can be observed for two parameters: the time spent eating and food intake. The HF animals tend to feed for a longer period and consume a higher amount of food compared to the NC animals.

In these tests, antagonistic behaviors indicating dominance over the other were considered (see methods in Section 2.3.7 and Section 2.3.8).

Like the previous test, the water/sucrose competition test is used to determine if there is a preference for drinking between sucrose and water. To carry this out, the amount of liquid ingested, the speed at which animals consume it, as well as the dominance of this resource among NC and HF animals were measured. When comparing significant differences within the same trial (when the liquid is the same), it was clear that, when they competed for water, males born from the HF mothers drank water for a longer time, although without significant differences in the amount of liquid intake. Note, also, that no differences were observed in the females in the time spent drinking water or the amount of water intake. Moreover, the animals, when exposed to water, did not exhibit significant differences in antagonistic behavior, the time spent displaying antagonistic behavior, and grooming. When the animals competed for sucrose, no significant differences were observed among the five parameters previously mentioned. When comparing the water trial vs. the sucrose trial, it seems that the animals spent more time consuming water than sucrose, exhibited a greater quantity of antagonistic behaviors when exposed to water compared to sucrose, and also engaged in a higher amount of grooming during the water trial.

### 3.4. Effect of Overnutrition during Pregnancy and Lactation on Hypothalamic, Hippocampal, and Prefrontal Markers of Synaptic Transmission, Metabolic Signaling, and Inflammation 

To assess the impact of excessive nutrition during pregnancy and lactation in some areas of the brain related with stress and anxiety—the hypothalamus, hippocampus, and prefrontal cortex—a Western blot technique was employed to examine the levels of some proteins and receptors involved in synaptic dynamics (Section 3.4.1), metabolism (Section 3.4.2), and inflammation (Section 3.4.3).

#### 3.4.1. Protein Markers of Synaptic Transmission on the Hypothalamus, Hippocampus, and Prefrontal Cortex 

Stress, anxiety, and cognition have been associated with altered synaptic transmission and plasticity [23,24]. To investigate synaptic dynamics in the three designated areas, we assessed three distinct proteins: the glutamate vesicular transporter (vGLUT), the synaptosome-associated protein 25 kDa (SNAP-25), which contributes to the SNARE complex involved in exocytosis, and the postsynaptic density protein 95 (PSD-95), a member of the membrane-associated guanylate kinase family that, with PSD-93, is recruited to the NMDA receptor and potassium channel clusters. 

In the hypothalamus, development induced increases of 37.5% and 44.7% (*p* < 0.001) in the levels of vGLUT, and the females showed a non-significant increase at PD28 and PD60. The male HF offspring did not exhibit significant differences compared to the NC offspring, but the female offspring at PD21 showed a significant increase of 57.5% that was attenuated with age (Figure 6A). Both the males and females showed increases in the SNAP-25 levels from PD21 onwards in the NC animals, with this being statistically significant only for the females at PD60 (increase of 99.3%). Moreover, both the males and females born from HF mothers exhibited altered levels of SNAP-25 at the hypothalamus, with the males always exhibiting higher levels of protein that were significant at PD28, and with the females exhibiting significant higher levels at PD21 that were maintained at PD28 and PD60 (Figure 6B). In relation to PSD-95, it was clear that the males exhibited a tendency to progressively increase the levels of this protein with age, which is an effect that was not observed in the females. Overnutrition during pregnancy and lactation did not change these results (Figure 6C). 

In the hippocampus, no significant differences were observed regarding age, diet, and sex for vGLUT (Figure 6D) or PSD-95 (Figure 6F). However, notable differences were evident in SNAP-25 in the males and females. Note that there was a clear decrease in the levels of SNAP-25 between PD21, PD28, and PD60 in the NC animals (males: decreases of 32.3% at PD28 and 52.6% at PD60; females: decreases of 52.2% at PD28 and 39.9% at PD60). Interestingly, the overfeeding of mothers during pregnancy and lactation did not change the levels of SNAP-25 in the males at PD21 and PD28, but it increased its levels by 37.5% at PD60. Moreover, females born from HF mothers showed significantly increased levels at PD21 and PD60 of 47.7% and 59.8%, respectively. 

The prefrontal cortex development in males and females produced significant changes in the levels of vGLUT, but not on the other proteins studied, where no differences were observed between PD21, PD28, and PD60 (Figure 6G–I). Note that the vGLUT levels were significantly higher at PD28 (increase of 70.3%) and PD60 (increase of 51.1%) in the males and at PD60 (increase of 37.4%%) in the females. Interestingly, only the male offspring from the HF mothers showed differences at PD21 that were attenuated with age (Figure 6G). The SNAP-25 levels were increased in the males and females born from HF mothers, an effect that was attenuated with age, but there was more marked in the males. Note that at PD21 and PD28, the males born from HF mothers showed high levels of SNAP-25 that were attenuated at PD60. Interestingly, this increase, although non-significant, was only seen in the females at PD21 (increase by 51.1%%) (Figure 6H). An HF diet during pregnancy and lactation did not change the levels of PSD-95 in the males nor females (Figure 6I).

#### 3.4.2. Protein Markers of Metabolism on the Hypothalamus, Hippocampus, and Prefrontal Cortex 

Figure 7 depicts the impact of hypercaloric diets consumed by mothers in some metabolic markers—insulin receptor and AMPK—in specific brain areas of the offspring. We can observe that in the hypothalamus, development significantly and progressively increased the levels of the ratio of IR-P/IR-T in the males and females born from NC mothers (males at PD28, 31.9% increase; males at PD60, 43.0% increase; females at PD28, 42.8% increase; females at PD60, 44.4% increase), suggesting a higher phosphorylation of the IR with development. Interestingly, while no changes were observed in the levels of this ratio at PD21 and PD28 in the males born from HF mothers, a significant reduction was seen at PD60 (49.3% decrease). In contrast, in the females, an HF diet in mothers promoted a higher ratio of IR-P/IR-T at PD21 (40.2% increase) that was attenuated with age (Figure 7A). Panel B shows the ratio of AMPK-P/AMPK-T, a metabolic sensor. Age increased the levels of the ratio of AMPK-P/AMPK-T in both the females and males, with these only being significant in the males (Figure 7B), suggesting a higher phosphorylation of AMPK with development. HF diet consumption in mothers did not change this ratio in neither sex (Figure 7B). In the hippocampus, the levels of the ratio of IR-P/IR-T followed, more or less, the same trend than in the hypothalamus, with the phosphorylation of IR increasing by 49.8% at PD60 in the males and increasing progressively with development in the females, although without statistical significance (Figure 7C). 

HF diet intake by mothers did not significantly alter the ratio of IR-P/IR-T in the offspring in neither sex (Figure 7C). Interestingly, we observed sexual dimorphism in the ratio of AMPK-P/AMPK-T at the hippocampus, with the females in the NC group showing a progressively and statistically significant increase in the phosphorylation of AMPK at PD28 (36.4%) and PD60 (36.3%), but no effect on the males was observed (Figure 7D). An HF diet in mothers significantly changed the ratio of AMPK-P/AMPK-T in the offspring in both sexes (Figure 7D). Figure 7E and F depict the levels of the same proteins in the prefrontal cortex. Note that there was a progressive decrease in the ratio of IR-P/IR-T with development in the offspring from NC mothers, which was more accentuated and statistically significant in the males (PD28, 38.8% decrease, and PD60, 55.4% decrease compared with PD21) than in the females (Figure 7E). Also note that HF diet intake in mothers promoted a decrease of 48.6% in the phosphorylation of IR at PD21 in the males that was not altered with age. In females, no differences were observed (Figure 7E). In contrast, the ratio of AMPK-P/AMPK-T statistically increased in the offspring of the NC mothers from PD21 onwards, both in the males (PD28m 13.6% increase; PD60, 15.4% increase) and females (PD28, 20.2% increase; PD60, 18.3% increase). HF diet intake in mothers induced a higher phosphorylation of AMPK in the offspring at PD21 in both the males (14.1% increase) and females (17.5% increase) that was maintained with age (Figure 7F).

#### 3.4.3. Protein Markers of Inflammation on the Hypothalamus, Hippocampus, and Prefrontal Cortex 

Inflammation has been implicated in several mood and anxiety disorders as well as in cognitive decline [25,26]. To explore the inflammation as being involved in the effects of overnutrition during pregnancy and lactation in stress and anxiety in the offspring, we evaluated the levels of three inflammatory markers: GFAP, whose expression was shown to be increased in brain inflammatory states [26]; TNF-α, a pro-inflammatory cytokine expressed in many brain pathologies and associated with neuronal loss [27]; and IL6-R, the receptor for IL6, a major cytokine that has anti- and pro-inflammatory effects on the brain and has been linked with neurodegenerative diseases and with chronic stress contributing to neurobehavioral complications [28].

In the hypothalamus, no significant changes were observed in the GFAP levels in males concerning diet or age. However, although the GFAP levels in the females did not change with age in the NC offspring, they exhibited a 38.5% significant increase from PD21 to PD28 in the HF offspring (Figure 8A). Surprisingly, the levels of TNF-α decreased from PD21 to PD28 and PD60 both in the males (PD28, 32.7%; PD60, 26.2%) and females (PD28, 36.4%; PD60, 52.4%) in the NC offspring. Overnutrition during pregnancy and lactation in mothers did not alter the levels of TNF-α in the hypothalamus in the offspring (Figure 8B). Considering IL6-R there was an increase in the levels of this protein with development in both the males (PD28, 36.4%; PD60, 91.8%) and females (PD28 32.5%) in the NC offspring. Overnutrition during pregnancy and lactation in the mothers did not significantly alter the levels of IL6-R in the hypothalamus in the offspring (Figure 8C). 

In the hippocampus, when examining GFAP, no significant differences were observed concerning age and diet for neither sex in the offspring (Figure 8D). In the analysis of the TNF-α, it became apparent that diet had a notable impact on the males. While there was a decrease in TNF with age in the male NC offspring, there was an increase in the levels of IL6-R in the HF offspring at PD28 and PD60 when compared to the NC group (35.8% and 51.2%, respectively), a trend that was not observed in the females (Figure 8E). When comparing the IL6-R levels in the males, an HF diet in the mothers increased the levels by 35.8% at PD21, which is an effect that was attenuated at PD28 and PD60. In females, age did not change the IL6-R levels in the NC offspring, but HF diet intake in mothers increased the IL6-R levels significantly by 28.1% at PD60 (Figure 8F). 

In the prefrontal cortex, the GFAP levels increased significantly with age in the males from the NC group (PD28, 39.6% and PD60, 32.5%), which is an effect that was absent in the females. Overnutrition during pregnancy and lactation in mothers did not significantly alter the levels of GFAP in the prefrontal cortex in the offspring (Figure 8G). TNF-α decreased with age in both the males and females born from NC mothers; however, the decrease was only statistically significant in the males (PD28, 41.5%; PD60, 47.5% in comparison with PD21). HF diet intake in mothers did not change the TNF-α levels in the males nor females (Figure 8H). In the analysis of the IL6-R levels, there was a 29.9% increase in the IL6-R levels at PD60 in the male offspring from NC mothers, while in the females, no alterations were observed with age. Overnutrition during pregnancy and lactation in mothers did not significantly alter the levels of IL6-R in the prefrontal cortex in the offspring (Figure 8I).

## 4. Discussion

In this study, we demonstrated that (1) female offspring born from overfeeding mothers during pregnancy and lactation showed increased weight gain and decreased glucose tolerance that attenuated with age; (2) offspring males born from overfeeding mothers exhibited increased weight gain that worsened with age, while glucose tolerance remained unchanged; (3) offspring from HF mothers exhibited increased levels of anxiety and stress during behavioral tests, displaying decreased predisposition for curiosity compared to the control group; and (4) offspring born from overfeeding mothers exhibited alterations in exocytotic capacity in the hypothalamus, hippocampus, and prefrontal cortex and in some inflammatory markers in the hippocampus that are different in males and females. As a whole, we demonstrated that maternal HF diet feeding during pregnancy and lactation induces dysmetabolism in the offspring accompanied by heightened stress and anxiety and by alterations in synaptic dynamics and neuroinflammation. Moreover, we found that there was sexual dimorphism in metabolic traits but not in behavior phenotypes.

### 4.1. Effect of Overnutrition during Pregnancy and Lactation on Metabolic Function in the Offspring

Herein, we showed, as expected, that throughout offspring development, distinctions arose concerning sex, diet, and age. As a pup develops, the body weight undergoes fluctuations depending on age, consistently showing a tendency to increase, and is largely influenced by dietary factors. Sexual dimorphism becomes apparent from 21 days onwards, with males generally exhibiting larger sizes and consequently greater weights than females. Over time, particularly in males, there is an escalating weight disparity influenced by diet. Conversely, in females, who commence hormone production at day 28 [29], the impact of diet on weight variation gradually diminishes. The estrogen hormone plays a pivotal role in regulating food intake and body weight, extending its influence to the modulation of insulin receptor abundance and ultimately being responsible for these differences between genders in overnutrition-dependent body weight gain [30].

Sexual dimorphism observed in the weight was also evident in the basal glycemia levels, with female offspring exhibiting changes in basal glycemia, while male offspring do not. This increased basal glycemia in female offspring at PD21 and PD28, compared to NC animals, suggests a significant impact of maternal diet during early development that could be attributed to a phenomenon known as developmental programming [31], where maternal influences during critical periods of fetal and neonatal development shape the long-term health of the offspring. Maternal overnutrition induces changes in the intrauterine environment, influencing the release of hormones and leading to epigenetic modifications [32]. These epigenetic changes can subsequently impact gene expression, potentially contributing to the observed differences in basal glycemia [33]. Interestingly, at PD60, the basal glycemia in female offspring from HF mothers did not differ significantly from the NC female offspring. This stabilization or resolution of metabolic effects over time suggests potential adaptive mechanisms or compensatory changes in response to early-life exposures and a possible contribution of hormonal effects [34]. Hormonal and genetic differences between male and female offspring may contribute to the observed variations in metabolic outcomes, and in this case, in basal glycemia [35].

Sexual dimorphism was also observed in insulin sensitivity. At PD21, the male offspring from HF mothers demonstrated a significant increase in the area under the curve of the insulin tolerance test, indicating decreased insulin sensitivity. This early effect suggests that maternal overnutrition during critical developmental periods may induce alterations in insulin signaling pathways, contributing to reduced insulin sensitivity in male offspring during early postnatal life. Surprisingly, at PD28, neither HF maternal feeding nor sex seemed to significantly impact the glycemia profiles during the ITT. This lack of effect at PD28 suggests potential adaptive responses or the normalization of insulin sensitivity in the offspring, possibly involving compensatory mechanisms that counteract the initial impact of maternal overnutrition [36]. By PD60, the male offspring, irrespective of maternal diet, displayed a significant increase in the area under the curve of the ipITT. This effect at PD60 suggests that the effects of age on insulin sensitivity that are clearly described in adulthood are already seen at early ages [37]. Regarding the female offspring, a different pattern emerged. The glycemia during ipITT indicated a significant increase in insulin sensitivity from PD21 to PD28 and a subsequent decrease from PD28 to PD60 in both the NC and HF female descendants. These dynamic changes suggest a time-dependent modulation of insulin sensitivity in females. Hormonal fluctuations, metabolic adaptation, or other sex-specific factors may contribute to these variations in insulin sensitivity during different developmental stages [37]. Moreover, the absence of effects of maternal dysmetabolism on insulin sensitivity with development contrasts with the documented impact of maternal obesity on insulin resistance later in life in animals [38] and humans [39]. However, this lack of effect on insulin sensitivity with development might be attributable to the duration of the exposure of mothers to the high-fat diet and their level of metabolic dysfunction. 

The glycemia profiles during the OGTT revealed that at each PD evaluated, neither the maternal diet nor offspring sex exerted a significant impact on glucose tolerance. This suggests a robustness or resilience in glucose tolerance to variations in maternal nutrition during the assessed developmental stages, indicating that the offspring can maintain a relatively stable glucose tolerance despite differences in the maternal diet or offspring sex [40]. However, a notable exception emerged in the female offspring at PD21. The female descendants of HF mothers exhibited a significant decrease in glucose tolerance compared to their counterparts from NC mothers. Interestingly, this effect in the female offspring was attenuated during development. The area under the curve of the OGTT decreased in the HF female offspring from PD21 to PD28 and remained relatively stable from PD28 to PD60. We have to take into account that females start hormonal production at PD28 [29]; therefore, at this age, there must be a potential adaptive response or compensatory mechanism in HF female offspring as they mature, indicating a dynamic regulation of glucose tolerance, with hormonal fluctuations in female development potentially having an important impact [40]. Conversely, the male offspring did not exhibit alterations in glucose tolerance during development based on the diet fed to the mothers. This lack of effect in males underscores a potential sex-specific response to maternal diet in the context of glucose tolerance during the assessed developmental period. Moreover, a recent meta-analysis showed that the exposure to maternal hyperglycemia during pregnancy might be associated with offspring obesity and abnormal glucose tolerance, although the association depends on the duration and intensity of intrauterine exposure to hyperglycemia [41], which agrees with the contrasting effects of our data with some of the published literature.

### 4.2. Effect of Overnutrition during Pregnancy and Lactation on Behavior and CNS Functions in the Offspring

In the present manuscript, we show that maternal exposure to an HF diet is correlated with a propensity for stress and anxiety-like behavior in the offspring without alterations in memory and learning and on food behavior. Our study demonstrates that offspring from HF mothers exhibited diminished exploratory behaviors and reduced curiosity, reflecting the same outcomes observed in previous studies involving offspring from HF mothers [42]. This behavioral trend is consistent with the existing literature that shows that rats subjected to an HF diet exhibit prolonged dwelling times in the shadowy corners of mazes and an increased frequency of defecation, aligning with the patterns reported by other researchers [43] and with human studies that demonstrated that prenatal maternal obesity is associated with offspring anxiety disorders, and that these associations may be long-lasting [44]. Moreover, it was shown that the severity of diabetes during pregnancy may increase the offspring’s vulnerability to depression/anxiety during childhood and adolescence [42], which is in agreement with our data presented herein. 

Trying to unveil the underlying neurobiological mechanisms by which maternal overnutrition induces stress and anxiety in the offspring, we focused on the synaptic dynamics and transmission, on metabolism, and on neuroinflammation. In the evaluation of synaptic dynamics, notably, neuroprotection was observed in the females, which could be primarily attributed to the presence of hormones, such as estrogen, which serve crucial roles in preserving and maintaining neuronal health [45]. This neuroprotective effect contributes to the manifestation of sexual dimorphism, particularly evident in the prefrontal cortex concerning markers like SNAP-25, a marker for exocytosis, and PSD-95, a marker for postsynaptic glutamatergic transmission. Additionally, the increase in vGLUT in the early days of life, which is also linked to glutamatergic transmission, may be associated with age and, once again, is influenced by the presence of hormones, especially in females, as noted in the hypothalamus and hippocampus [44]. Regarding the impact of overnutrition in mothers in the offspring, this is mainly observed at the exocytosis level, with HF-exposed pups displaying an increased rate of exocytosis in the three regions studied—the hypothalamus, hippocampus, and prefrontal cortex. However, we can also see a sexual dimorphism in synaptic transmission since the increase in the exocytotic marker, SNAP-25, seen in males was not observed in females neither in the hypothalamus nor prefrontal cortex, and it was even decreased at PD60 in the hippocampus in females. Conversely, in the post-synaptic region, no discernible changes were evident with respect to either diet or age on the glutamatergic marker. While there is some evidence reporting altered glutamatergic signaling in the amygdala in the offspring from obese mothers [45,46], in the three regions studied in the present manuscript, overnutrition in mothers clearly modifies markers of glutamatergic signaling, suggesting that the effects of a high fat diet involve neurotransmitters other than glutamate. This agrees with some demonstrations of decreased levels of GABAergic and serotonergic neurotransmitters in the whole brain [47]. Moreover, since hypothalamic neurotransmitters like POMC and NPY are critical for energy balance and feeding [48] (it was found that a POMC-originated circuit regulates stress [49], the ablation of POMC neurons led to anxiety-like behavior [50], and NPY knockouts are related to anxiety [51]), other neurotransmitters will probably be keys in driving this stress/anxiety phenotype in the offspring of dysmetabolic mothers. As a whole, our results clearly suggest a selective influence of dietary factors on specific aspects of synaptic function, underscoring the intricate interplay between nutrition, gender-related hormonal influences, and synaptic mechanisms.

Concerning metabolism molecular pathways, an intriguing pattern emerges as males exhibit an age-related increase in the activation of the insulin signaling pathway, while females maintain a consistent level throughout the early days of development. Notably, in the prefrontal cortex of males, there is a decline in insulin receptor phosphorylation with age. This trend suggests a plausible scenario wherein the initial higher activation of these receptors to participate in the high-rate metabolism at birth diminishes as the pup matures [46,47,48]. Interestingly, no significant effect of HF diets in mothers was appreciated in the insulin signaling cascade in the offspring except for the statistically significant decrease in activation at PD60 in the hypothalamus in males. Note that, at the hypothalamus, insulin suppresses food intake, is involved in glucose and fat metabolism regulation [52,53], and controls sympathetic activity [54]. Therefore, a decreased activation of this pathway at PD60 can anticipate the late development of cardiometabolic complications. Regarding AMP-activated protein kinase (AMPK), the activation of this protein experiences an increase across various brain regions and age groups. This versatile protein not only facilitates the conversion of AMP into ATP but also responds to nutrients, metabolites, and hormones involved in energy balance [55], and it is also involved in the regulation of growth and reprogramming metabolism [56], which could explain the increase during development. Herein, we observed slight increases in AMPK signaling at PD21 in the offspring of HF mothers, both in males and females, that were attenuated with age. Within the hormones that modulate AMPK-dependent metabolic control, there is insulin, leptin, ghrelin, estrogens, etc. Therefore, hormonal maturation is a crucial step that will change AMPK pathway activation in the control of food intake and energy metabolism. Moreover, the distinct hormonal maturation between sexes will make AMPK modulation different in males and females due to the roles of sex-specific hormones that were previously described [54]. Given its multifaceted role in various pathways, further investigations are warranted to pinpoint the specific pathways activated by AMPK and its specific role in the contexts of brain function [49,50] and the transgenerationality of dysmetabolism.

When the inflammatory markers were analyzed, we showed a general trend where the females exhibited lower levels of these markers compared to the males. This distinction is notably apparent from the age of 28 days onward, coinciding with the onset of hormone production in females, which may act as a protective mechanism against external stressors, thereby mitigating the risk of elevated inflammation [57]. Note also that in contrast, males, on average, display an upswing in inflammatory markers, namely GFAP and IL6-R, when they are descendants of NC mothers. Notably, all of the regions studied in the present manuscript exhibited a higher amount of the TNF-α molecule at PD21 that decreased with development. Throughout uterine development, pups are constantly protected from external agents, benefiting from the protective environment provided by their mother’s uterus, including a regulated temperature. Consequently, it is expected that their inflammatory levels remain low during this stage of development [58]. However, after birth, the newborn is exposed to a variety of external agents and environmental conditions that can trigger an initial increase in inflammatory levels during the first few days [59]. Afterwards, the newborn gradually acclimatizes to these external factors, leading to a reduction in inflammation levels [60]. The tendency for inflammation levels to decrease with age reflects the continuous acclimatization of the newborn to the external environment. Regarding the effect of the overnutrition of mothers on the offspring while no alterations in the GFAP levels, which have been previously associated with neuroinflammation [25,26], we observed increased levels of TNF-α, particularly on the hippocampus of the males, and increased levels of IL6-R at the hypothalamus and at the hippocampus in both males and females, suggesting that neuroinflammation in a neurobiological process underlies the development of stress and anxiety behaviors. This agrees with previous studies that demonstrated that anxiety-like behavior was associated with the increased mRNA expression of proinflammatory markers, including IL6, TNFα, NFkB, and MCP-1, in the hypothalamus and amygdala [61,62,63]. However, more information will be needed, and more markers of inflammation should be tested and in different models (different exposure times to diets and different diets) to define an association between neuroinflammation and stress and anxiety behaviors in the offspring.

## 5. Conclusions

In conclusion, this work demonstrates that, as shown before, exposure to an HF diet during pregnancy and lactation induces dysmetabolism in the offspring and adds information related with the behavioral impact of maternal overnutrition, showing that it induces heightened stress and anxiety. Furthermore, it unequivocally demonstrates that increased stress and anxiety are correlated with changes in synaptic dynamics and neuroinflammation in the hypothalamus, hippocampus, and prefrontal cortex. Finally, we showed that most of these effects of maternal overnutrition during pregnancy and lactation show sexual dimorphism in metabolic traits but not in behavioral phenotypes.

## Figures and Tables

**Figure 1 nutrients-16-00067-f001:**
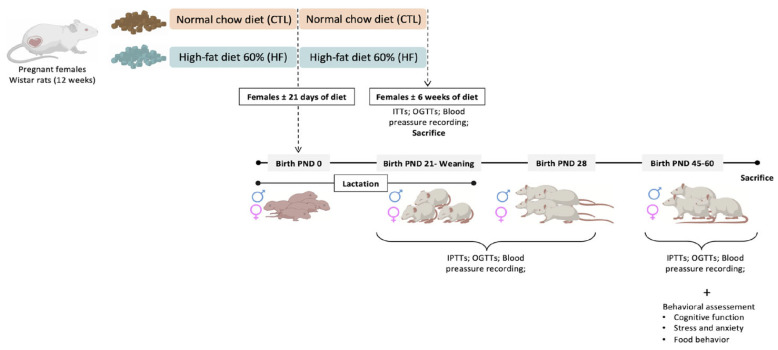
Schematic illustration of the protocol of the study. CTL—control; HF—High-fat diet; PND—post-natal day. 
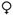
—female; 
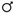
—male.

**Figure 2 nutrients-16-00067-f002:**
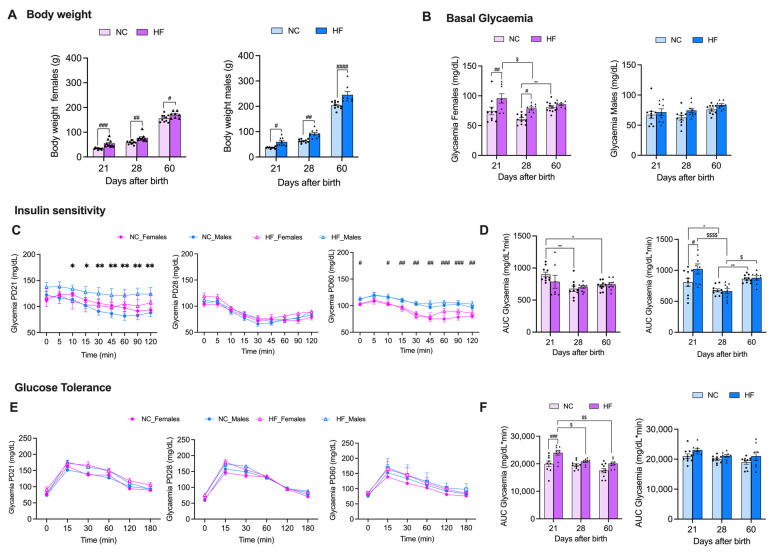
Effect of overnutrition during pregnancy and lactation on metabolic phenotype of the offspring, evaluated by body weight gain (**A**), basal glycemia (**B**), insulin sensitivity (**C**,**D**), and glucose tolerance (**E**,**F**) at postnatal days 21 (PD21), 28 (PD28), and 60 (PD60). (**A**) From the left to the right, body weight in g during offspring development for females and males. (**B**) From the left to the right, basal glycemia in mg/dL during early development for females and males. (**C**,**D**) Insulin sensitivity represented by the glycemia profiles during ipITT (**C**) and by the area under the curve of those glycemia curves during early descendants’ development for females and males, from the left to the right, respectively (**D**). (**E**,**F**) Glucose tolerance represented by glycemia profiles during OGTT (**E**) and by the area under the curve of those glycemia curves during early descendants’ development for females and males, from the left to the right, respectively (**F**). PD21—postnatal day 21; PD28—postnatal day 28; PD60—postnatal day 60. Data represent mean ± SEM of 10 NC females, 10 NC males, 9 HF females, and 9 HF males per time-point. Black circles correspond to offspring born form mothers submitted to NC diet and black triangles to offspring born form mothers submitted to HF diet. Two-way ANOVA with Bonferroni and Sidak’s multiple comparison tests; * *p* < 0.05 and ** *p* < 0.01 comparing NC animals with different ages; ^#^ *p* < 0.05, ^##^ *p* < 0.01, ^###^ *p* < 0.001 and ^####^ *p* < 0.0001 comparing NC vs. HF animals with the same age; and ^$^ *p* < 0.05, ^$$^ *p* < 0.01, and ^$$$$^ *p* < 0.0001 comparing HF animals with different ages.

**Figure 3 nutrients-16-00067-f003:**
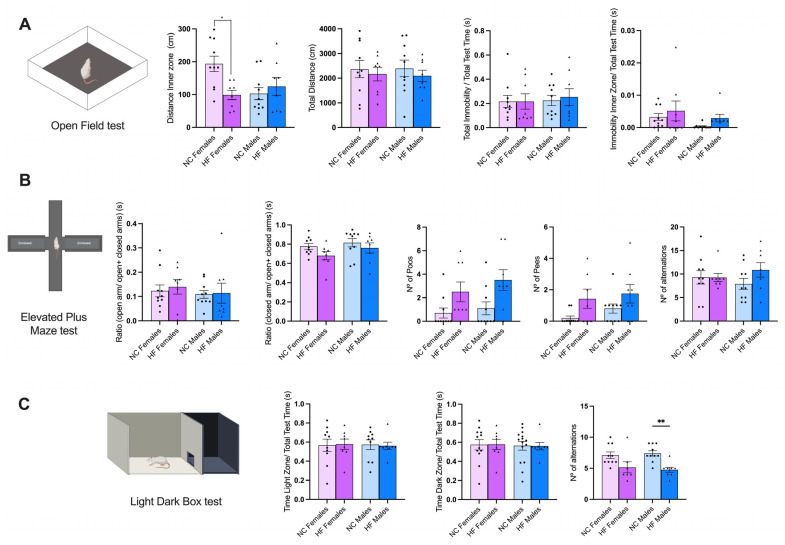
Effect of overnutrition during pregnancy and lactation on anxiety and stress, evaluated using the open field test (**A**), elevated plus maze test (**B**), and light–dark box test (**C**), in the offspring at postnatal day 60. (**A**) From the left to the right, the panel shows a schematic representation of the open field maze, the distance in the inner zone (cm), total distance covered (cm), total immobility (s), and the immobility in the inner zone (s); (**B**) from left to right, the panel shows a schematic representation of the elevated plus maze, the time spent in the open arms in comparison to the time spent in all arms (s), the time spent in the closed arms in comparison to the time spent in all arms (s), the number of poops in the maze, the number of pees, and the number of alternations between the open arm and the closed arms. Data represent mean ± SEM of 10 NC females, 10 NC males, 8 HF females, and 8 HF males. Black circles correspond to offspring born form mothers submitted to NC diet and black triangles to offspring born form mothers submitted to HF diet. Two-way ANOVA with Bonferroni and Sidak’s multiple comparison tests were used; * *p* < 0.05 and ** *p* < 0.01 comparing NC animals.

**Figure 4 nutrients-16-00067-f004:**
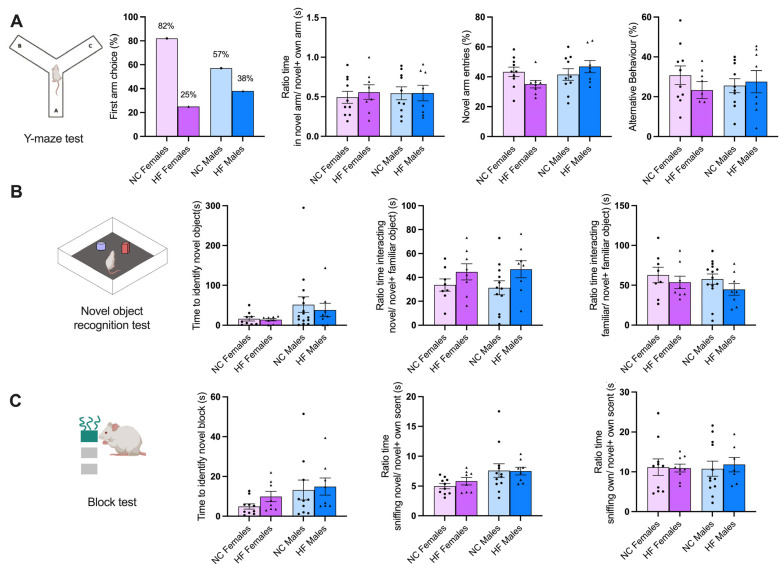
Effect of overnutrition during pregnancy and lactation on memory and learning, evaluated using the Y maze test (**A**), novel object recognition test (**B**), and block test (**C**), in the offspring at postnatal day 60. (**A**) From the left to the right, panel shows a schematic representation of the Y maze, the first arm choice (%), the ratio between the novel arm vs. the time in both the novel and familiar arm, the % of novel arm entries, and the alternative behaviors (%). (**B**) From the left to the right, the panel presents a schematic representation of novel object recognition apparatus, the time taken to identify the novel object (s), the ratio of the time the animals spent interacting with the novel object vs. the total interaction time (s), and the time the animals spent interacting with the familiar object vs. the total interaction time (s). (**C**) From left to right, the panel presents a schematic representation of the block test, the time taken to identify the novel block (s), the time the animals spent sniffing the novel block vs. the total time sniffing all the blocks, and the time the animals spent sniffing the familiar blocks vs. the total time sniffing all of the blocks. Data represent mean ± SEM of 10 NC females, 10 NC males, 8 HF females, and 8 HF males. Black circles correspond to offspring born form mothers submitted to NC diet and black triangles to offspring born form mothers submitted to HF diet. Two-way ANOVA with Bonferroni and Sidak’s multiple comparison tests were conducted.

**Figure 5 nutrients-16-00067-f005:**
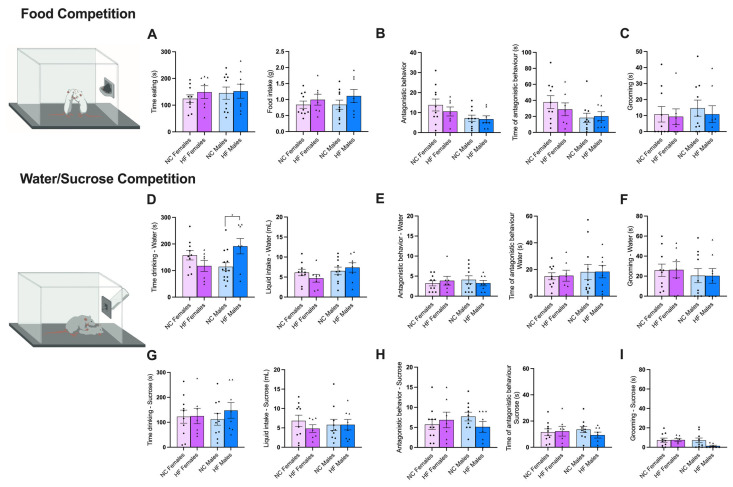
Effect of overnutrition during pregnancy and lactation on food behavior, evaluated using food competition test (**A**–**C**) and water/sucrose competition test (**D**–**I**). (**A**) From the left to the right, the panel shows a schematic representation of the food competition test, the time spent eating (s) and the food intake (g) (**A**), the antagonistic behavior and the time spent displaying antagonistic behavior (s) (**B**), and the time of grooming (s) (**C**). (**D**–**F**) From the left to the right, the panel represents a schematic representation of water/sucrose competition, time drinking (s) and the liquid intake (mL) (**D**), the antagonistic behavior and the time spent displaying antagonistic behavior (s) (**E**), and time of grooming (s) (**F**) in response to water competition. (**G**–**I**) From the left to the right, the panel shows the time drinking (s) and the liquid intake (mL) (**G**), the antagonistic behavior and the time spent displaying antagonistic behavior (s) (**H**), and time of grooming (s) (**I**) in the presence of sucrose. Data represent mean ± SEM of 10 NC females, 10 NC males, 8 HF females, and 8 HF males. Black circles correspond to offspring born form mothers submitted to NC diet and black triangles to offspring born form mothers submitted to HF diet. Two-way ANOVA with Bonferroni and Sidak’s multiple comparison test were conducted; * *p* < 0.05 comparing NC animals.

**Figure 6 nutrients-16-00067-f006:**
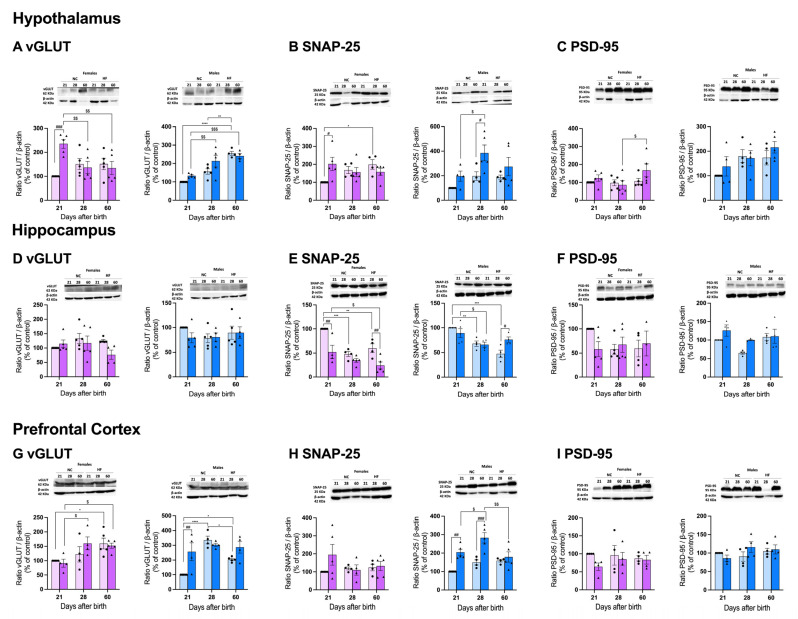
Effect of overnutrition during pregnancy and lactation on the levels of proteins involved in synaptic transmission on the hypothalamus (**A**–**C**), hippocampus (**D**–**F**), and prefrontal cortex (**G**–**I**). From the left to the right, graphs show the mean levels of the ratio between vGLUT and β-actin, the ratio between SNAP-25 and β-actin, and the ratio between PSD-95 and β-actin in males and females, respectively at the hypothalamus (**A**–**C**), hippocampus (**D**–**F**), and prefrontal cortex (**G**–**I**). On the top of the graphs are representative Western blot membranes for the proteins of interest and the respective loading control. Data represent mean ± SEM of 4–5 NC females, 4–5 NC males, 4–5 HF females, and 4–5 HF males for each time-point. Black circles correspond to offspring born form mothers submitted to NC diet and black triangles to offspring born form mothers submitted to HF diet. Two-way ANOVA with Tukey’s and Sidak’s multiple comparison tests were conducted; * *p* < 0.05, ** *p* < 0.01, *** *p* < 0.001 and **** *p* < 0.0001 comparing NC animals with different ages; ^#^ *p* < 0.05, ^##^ *p* < 0.01, and ^###^ *p* < 0.001 comparing NC vs. HF animals with the same age; and ^$^ *p* < 0.05, ^$$^ *p* < 0.01 and ^$$$^ *p* < 0.001 comparing HF animals with different ages.

**Figure 7 nutrients-16-00067-f007:**
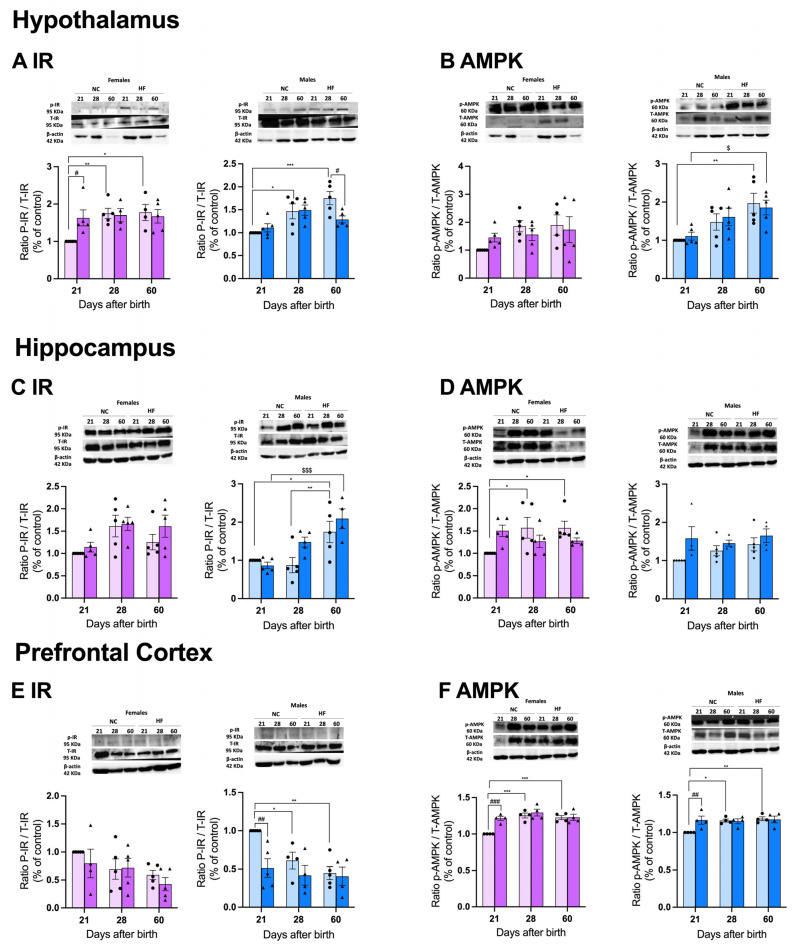
Effect of overnutrition during pregnancy and lactation on the levels of insulin receptor and AMPK on the hypothalamus (**A**,**B**), hippocampus (**C**,**D**), and prefrontal cortex (**E**,**F**). From the left to the right, graphs represent mean levels of the ratio between phosphorylated insulin receptor/total insulin receptor (P-IR/T-IR) and β-actin, and the ratio between phosphorylated AMPK and total AMPK (p-AMPK/T-AMPK) and β-actin in males and females, respectively at the hypothalamus (**A**,**B**), hippocampus (**C**,**D**), and prefrontal cortex (**E**,**F**). On the top of the graphs are representative Western blot membranes for the proteins of interest and the respective loading control. Data represent mean ± SEM of 4–5 NC females, 4–5 NC males, 4–5 HF females, and 4–5 HF males for each time-point. Black circles correspond to offspring born form mothers submitted to NC diet and black triangles to offspring born form mothers submitted to HF diet. Two-way ANOVA with Tukey’s and Sidak’s multiple comparison tests were conducted; * *p* < 0.05, ** *p* < 0.01 and *** *p* < 0.001 comparing NC animals with different ages; ^#^ *p* < 0.05, ^##^ *p* < 0.01, and ^###^ *p* < 0.001 comparing NC vs. HF animals with the same age; and *^$^ p* < 0.05, *^$$$^ p* < 0.001 comparing HF animals with different ages.

**Figure 8 nutrients-16-00067-f008:**
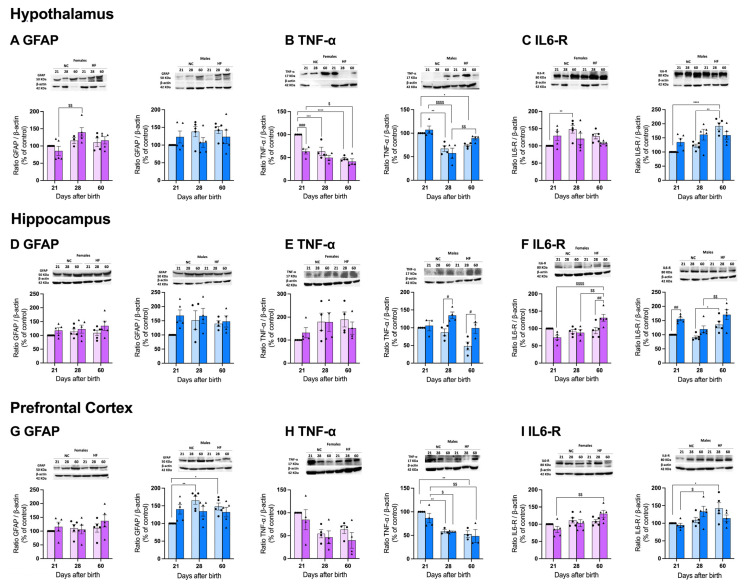
Effect of overnutrition during pregnancy and lactation on the levels of proteins involved in inflammation on the hypothalamus (**A**–**C**), hippocampus (**D**–**F**), and prefrontal cortex (**G**–**I**). From the left to the right, graphs show mean levels of the ratio between GFAP and β-actin, the ratio between TNF-α and β-actin, and the ratio between IL6-R in males and females, respectively at the hypothalamus (**A**,**B**), hippocampus (**C**,**D**), and prefrontal cortex (**E**,**F**). On the top of the graphs are representative Western blot membranes for the proteins of interest and the respective loading control. Data represent mean ± SEM of 4–5 NC females, 4–5 NC males, 4–5 HF females, and 4–5 HF males for each time-point. Black circles correspond to offspring born form mothers submitted to NC diet and black triangles to offspring born form mothers submitted to HF diet. Two-way ANOVA with Tukey’s and Sidak’s multiple comparison tests were conducted; * *p* < 0.05, ** *p* < 0.01, *** *p* < 0.001 and **** *p* < 0.0001 comparing NC animals with different ages; ^#^ *p* < 0.05, ^##^ *p* < 0.01, and ^###^ *p* < 0.001 comparing NC vs. HF animals with the same age; and ^$^ *p* < 0.05, ^$$^ *p* < 0.01, and ^$$$$^ *p* < 0.0001 comparing HF animals with different ages.

**Table 1 nutrients-16-00067-t001:** Impact of high-fat diet on metabolic parameters in mothers.

	NC Mothers	HF Mothers
Body Weight (g)	243.2 ± 5.122	236.8 ± 2.577
Basal Glycaemia (mg/dL)	69.14 ± 1.792	78.88 ± 3.573 *
k_ITT_ (% glucose/min)	4.864 ± 0.3957	1.849 ± 0.3188 ****
AUC Glycaemia (mg/dL·min)	18,138 ± 417.4	19,612 ± 594.9
Liver (g/kg)	0.0280 ± 0.001	0.0289 ± 0.001
Visceral Adipose Tissue (g/kg)	0.0057 ± 0.002	0.0066 ± 0.002
Perinephric Adipose Tissue (g/kg)	0.0153 ± 0.002	0.0171 ± 0.004
Genital Adipose Tissue (g/kg)	0.0232 ± 0.002	0.0254 ± 0.002
Brown Adipose Tissue (g/kg)	0.0007 ± 0.000	0.0016 ± 0.000

Data represent mean ± SEM of 5 NC mothers and 5 HF mothers. * *p* < 0.05, **** *p* < 0.0001. Student’s *t* test was used to compare values in animals submitted to normal chow (NC) and animals submitted to high-fat diet (HF). K_ITT_—constant of the insulin tolerance test; AUC—area under the curve of the glucose excursion curves obtained from the oral glucose tolerance test.

**Table 2 nutrients-16-00067-t002:** Impact of overnutrition in mothers during pregnancy and lactation on the weights (mg) of the liver and of the different adipose tissue (AT) depots in male and female descendants at postnatal days 21, 28, and 60.

	Liver	Visceral AT	Perinephric AT	Genital AT	Brown AT
NC Females PD21	0.963 ± 0.066	0.262 ± 0.026	0.105 ± 0.017	0.128 ± 0.015	0.087 ± 0.010
HF Females PD21	1.646 ± 0.139	0.278 ± 0.039	0.222 ± 0.026	0.318 ± 0.098	0.130 ± 0.021
NC Females PD28	0.978 ± 0.056	0.257 ± 0.021	0.139 ± 0.012	0.114 ± 0.012	0.103 ± 0.009
HF Females PD28	1.965 ± 0.115	0.344 ± 0.042	0.303 ± 0.053	0.286 ± 0.050	0.143 ± 0.015
NC Females PD60	1.892 ± 0.118	0.406 ± 0.038	0.196 ± 0.017	0.214 ± 0.033	0.135 ± 0.009
HF Females PD60	2.670 ± 0.158	0.527 ± 0.055 ***	0.374 ± 0.068 **	0.472 ± 0.075	0.139 ± 0.010
NC Males PD21	2.286 ± 0.086	0.376 ± 0.042	0.376 ± 0.059	0.305 ± 0.033	0.146 ± 0.016
HF Males PD21	2.717 ± 0.126 **	0.561 ± 0.043	0.472 ± 0.042	0.528 ± 0.054	0.144 ± 0.012
NC Males PD28	4.872 ± 0.405	1.029 ± 0.135	1.109 ± 0.212	1.993 ± 0.405	0.249 ± 0.025
HF Males PD28	4.566 ± 0.178	1.472 ± 0.065	1.879 ± 0.281	2.183 ± 0.386	0.226 ± 0.026
NC Males PD60	6.377 ± 0.326	1.216 ± 0.182	2.599 ± 0.381	2.324 ± 0.375	0.269 ± 0.025
HF Males PD60	7.218 ± 0.465 *	1.864 ± 0.152 **	3.162 ± 0.422	3.642 ± 0.503 **	0.284 ± 0.026

Data represent mean ± SEM of 10 NC females, 10 NC males, 9 HF females, and 9 HF males per time-point.* *p* < 0.05, ** *p* < 0.01; *** *p* < 0.001 two-way ANOVA with Sidak’s multiple comparison test comparing animals submitted to normal chow (NC) diet and animals submitted to high-fat diet (HF) of the same age. PD21—postnatal day 21; PD28—postnatal day 28; PD60—postnatal day 60.

## Data Availability

The data that support the findings of this study are available from the corresponding author upon request.

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
