# Peer review of "Overnutrition during Pregnancy and Lactation Induces Gender-Dependent Dysmetabolism in the Offspring Accompanied by Heightened Stress and Anxiety"

_nutrients, 2023, doi:10.3390/nu16010067_

Round 1

Reviewer 1 Report

Comments and Suggestions for Authors

Overall this study was well performed and correctly interpretated- but as written it is not presented in a clear and concise manner.  A couple minor point to address:

1-Table 1: kITT Neds defined in the legend.  

2-Figure 1: Where is figure 1?  Or is figure 2 mislabeled?

3-Figure 2a/b/c/d bar graphs: significance lines are offset and do not line up with bars. Additionally figure B, C2, and D2 are hard to interpret as drawn.  Suggest dropping lines to indicate bars in question

4-Figure 2C1 / C2 and D1 / D2:  this is a strange way to present these.  Suggest sticking with letters only (C,D,E,and F)

5-lines 341-344: This is impossible to follow with the short hand and acronyms.  Suggest either referring to the graph or writing in normal sentence structure

6-Table 2: the split legend makes this table hard to follow.  The legend should be either completely below or completely above the figure.  Also the units should be in the table headings themselves.

7-Figure 4b and lines 433-434: To claim the data is trending you must include a p value approaching significance.  None is found.

8-Figure 5: see previous comment on unconventional labeling.  Additionally the y axis on B1, B2, B5, and B6 are well over the maximum value presented resulting in little information to be able to gained from the graphs.

9-Figures 6/7/8:  again labeling concerns.  Also the blots are too small to interpret without extensive zooming in.  They should be visible on a printout.  This is an extensive amount of work, you should display it clearly

Comments on the Quality of English Language

minor grammar issues, can be corrected by editor

Author Response

Overall this study was well performed and correctly interpretated- but as written it is not presented in a clear and concise manner.  A couple minor point to address:

We appreciate the reviewer comments about our work. We have reviewed the manuscript according with the reviewers’ suggestions and hope that in the present form the manuscript is suitable for publication.

1-Table 1: kITT Neds defined in the legend.  – done as suggested.

2-Figure 1: Where is figure 1?  Or is figure 2 mislabeled? – Figure 1 is the schematic illustration of the protocol of the study and is referenced in the line 155 of the manuscript.

3-Figure 2a/b/c/d bar graphs: significance lines are offset and do not line up with bars. Additionally figure B, C2, and D2 are hard to interpret as drawn.  Suggest dropping lines to indicate bars in question. Altered as suggested.

4-Figure 2C1 / C2 and D1 / D2:  this is a strange way to present these.  Suggest sticking with letters only (C,D,E,and F). Altered as suggested.

5-lines 341-344: This is impossible to follow with the short hand and acronyms.  Suggest either referring to the graph or writing in normal sentence structure. We understand that there are a lot of acronyms – KITT, AUC and PD - that refers to the constant of the insulin tolerance test, the area under the curve of the OGTT and post-natal day, however from our point of view it can be important to the readers to have access to the absolute values of the metabolic parameters.

6-Table 2: the split legend makes this table hard to follow.  The legend should be either completely below or completely above the figure.  Also the units should be in the table headings themselves. Altered as suggested.

7-Figure 4b and lines 433-434: To claim the data is trending you must include a p value approaching significance.  None is found. We have removed the sentence related with this data.

8-Figure 5: see previous comment on unconventional labeling.  Additionally the y axis on B1, B2, B5, and B6 are well over the maximum value presented resulting in little information to be able to gained from the graphs. We have altered as suggested by the reviewer.

9-Figures 6/7/8:  again labeling concerns.  Also the blots are too small to interpret without extensive zooming in.  They should be visible on a printout.  This is an extensive amount of work, you should display it clearly. We have altered the labelling as suggested by the reviewer. Also, we have increased “as possible” the membranes of the western blots and the graphs.

Reviewer 2 Report

Comments and Suggestions for Authors

Overall, I think the paper has a very good quality (and it is well situated within the scope of this Journal). The manuscript addresses an interesting issue about the role of maternal obesity, lactation and gestational diabetes predispose to metabolic disturbances in the next generation. Moreover, the lactation phase also stands as critical for metabolic programming. This study investigates the consequences of maternal diet lipid-rich diet during gestation and lactation, and its impact on metabolism and behavior in the offspring.

The manuscript provides a substantial original contribution to the scientific literature and can be interesting to a wide range of professional readers. Moreover, the manuscript effectively conveys the main findings, it is understandable, well-structured and all sections are interesting and relevant. It provides a good amount of information to support the arguments being made and the methods used are correct. The discussion is well written and the conclusions of the paper are clearly stated and adequately tie together the other elements of the manuscript, although should be increased to increase its strength.

The style and grammar are in general correct, although I have some comments which I think could improve the manuscript:

-          The introduction should be increased in order to highlight the role of maternal obesity on the hypothalamic role oof hunger/statiety on the offspring and its relationship with endocrine, metabolic, and inflammatory processes conditioning the postnatal life, to support the discussion.

-          Material and methods:

A) the number of female, male Wistar and pups of each group must be cleraly stated in the experimental design, the authors only mention 8-10 animals per group in the statistical analysis.

B) Was the diet available ad libitum of pair feeding? This also has to be stated and in case of pair feeding, how was the amount of diet supplied calculated?

-          The conclusions of the paper are clearly stated, although they are scarcely described. I encourage the authors to include interesting findings such as the effect of overnutrition during pregnancy on metabolic parameters and their relationships with postnatal development and the expression of protein markers of inflammation on the hypothalamus, hippocampus and prefrontal cortex to increase the strength of the conclusions.

Author Response

We acknowledge the reviewer the positive comments of our work. We have followed your suggestions and hope that in the present form the manuscript is suitable for publication.

The style and grammar are in general correct, although I have some comments which I think could improve the manuscript: - The introduction should be increased in order to highlight the role of maternal obesity on the hypothalamic role oof hunger/statiety on the offspring and its relationship with endocrine, metabolic, and inflammatory processes conditioning the postnatal life, to support the discussion. We have now included a paragraph in the introduction dealing with the effects of maternal obesity in the hypothalamic role of hunger/satiety on the offspring and its relationship with endocrine, metabolic, and inflammatory processes conditioning the postnatal life to support discussion. Now it can be read “Studies in animals have shown that maternal obesity can increase the likelihood of metabolic and neurodevelopmental diseases in the offspring, such as cognitive impairment, autism spectrum disorders and attention deficit hyperactivity disorder, [2-4]. Moreover, maternal obesity and dysmetabolism are linked to a heightened risk of psychiatric disorders in offspring, encompassing conditions such as anxiety, depression, schizophrenia, psychosis, eating disorders, and food addiction (see e.g ref). Several mechanisms have been postulated to underlie these neurodevelopment and psychiatric conditions including oxidative stress and inflammation-induced malprogramming, the deregulation of insulin, glucose, and leptin signaling, the dysregulation synaptic transmission or even the deregulation of the gut-brain axis nevertheless, the precise ones are not completely understood.”.

Material and methods:

A) the number of female, male Wistar and pups of each group must be clearly stated in the experimental design, the authors only mention 8-10 animals per group in the statistical analysis. We have included the number of animals in the legend of all figures and tables.

B) Was the diet available ad libitum of pair feeding? This also has to be stated and in case of pair feeding, how was the amount of diet supplied calculated? Diet was available ad libitum as stated in methods section in line 113.

- The conclusions of the paper are clearly stated, although they are scarcely described. I encourage the authors to include interesting findings such as the effect of overnutrition during pregnancy on metabolic parameters and their relationships with postnatal development and the expression of protein markers of inflammation on the hypothalamus, hippocampus and prefrontal cortex to increase the strength of the conclusions. We have made a summary of the main conclusions in the conclusion section of the manuscript.